# The Eating Healthy and Daily Life Activities (EHDLA) Study

**DOI:** 10.3390/children9030370

**Published:** 2022-03-07

**Authors:** José Francisco López-Gil

**Affiliations:** 1Departamento de Expresión Plástica, Musical y Dinámica, Facultad de Educación, Universidad de Murcia (UM), 30100 Murcia, Spain; josefranciscolopezgill@gmail.com; 2Health and Social Research Center, Universidad de Castilla-La Mancha (UCLM), 16071 Cuenca, Spain

**Keywords:** obesity, excess weight, physical activity, sedentary behavior, sleep duration, eating healthy, lifestyle, psychological aspects, youths, adolescence

## Abstract

Background: Childhood obesity is one of the greatest public health concerns facing advanced societies, Spain being one of the countries with the highest incidence. In this sense, the Region of Murcia has been pointed out as the Spanish autonomous community with the highest prevalence of excess weight among young people. More specifically, the Valle de Ricote has shown an even greater proportion of excess weight among young people. Several sociodemographic, environmental, lifestyle, health-related, cognitive, and psychological factors are related to excess weight. Based on the lack of information, this research project will try to provide relevant information to design intervention programs, as well as to implement effective public policies to try and reverse this alarming situation. Therefore, this research project aims (1) to obtain cross-sectional and longitudinal data on the excess weight and their potential sociodemographic, environmental, lifestyle, health-related, cognitive, and psychological factors associated among adolescents from the Valle de Ricote (Region of Murcia, Spain) (aged 12–17 years), and (2) to examine the association between excess weight and their potential sociodemographic, environmental, lifestyle, health-related, cognitive, and psychological factors associated among this population. Methods: A cross-sectional study and follow-up study will be performed. This research project will involve adolescents using a simple random sampling technique. A total of three secondary schools from the Valle de Ricote (Region of Murcia, Spain) will be included in this project. The minimum sample size will be 1138, establishing a 95% confidence interval, a 40% prevalence of excess weight, a 3% margin of error, and a non-response rate of 10%. Primary outcome measures will be: (1) anthropometric measurements, (2) sociodemographic factors, (3) environmental factors, (4) lifestyle factors, (5) health-related factors, (6) cognitive factors, and (7) psychological factors. Conclusion: This research project will aim to determine the prevalence of excess weight and interrelate their potential sociodemographic, environmental, lifestyle, health-related, cognitive, and psychological factors associated. The obtained results will help to manage and propose possible multidisciplinary interventions and strategies in order to prevent and reduce the excess weight in adolescents from the Valle de Ricote. Furthermore, orientations will be given to transfer the obtained results to the public sector to evaluate or change the adopted policies.

## 1. Introduction

The burden of excess weight (overweight and obesity) has reached worryingly high levels globally [1], especially in Europe, where this situation represents a major public health threat [2]. The last World Health Organization (WHO) European Childhood Obesity Surveillance (COSI) study ranked Spain as having the highest obesity prevalence rate in Europe (17.7%) [2]. However, excess weight prevalence is not homogenous across all autonomous communities in Spain, since it differs according to the different regions [3]. In this sense, the highest prevalence used to be in the southernmost regions of the country [3,4]. This is worrisome since the risk of morbidity and mortality in adult life increases in those who have excess weight in their childhood or adolescence [5], which also increases the likelihood of having higher values of risk parameters for selected adult cardiovascular disease risk factors [6]. This is why the WHO Member States have committed to warrant that excess weight among young population does not increase by 2025 [7].

Within the Region of Murcia, several studies have reported a high prevalence/proportion of young population with excess weight. For instance, the last Spanish National Health Survey (2017) indicated that the young population between 2 and 17 years of age with excess weight stood at 40.0%, 28.0% being in the Spanish territory [8]. Similarly, a recent longitudinal study tracking excess weight from 2005 to 2017 found that the Region of Murcia is the autonomous community with the highest obesity prevalence in Spain [9]. That is why the United Nations International Children’s Emergency Fund (UNICEF) has pointed out with a red flag this worrying situation about childhood obesity in this autonomous community [10]. Regarding the Valle de Ricote (Region of Murcia), one study performed in this area found a proportion of 52.4% of excess weight (according to WHO criteria) among schoolchildren [11].

Several sociodemographic factors are related to excess weight. For instance, one cohort study among Spanish youth showed that sex, age, and socioeconomic and immigrant status were related to excess weight [9]. Environmental factors are related to excess weight, such as family home environment [12], physical environment [13], or barriers to physical activity practice [14]. Similarly, several lifestyle factors (e.g., movement behavior, eating patterns or drug consumption) are associated with excess weight among children and adolescents, such as physical activity, eating breakfast, fruits/vegetables intake, inadequate sleep duration, watching TV, playing computer games, eating sweets, eating snack, drinking sugar-sweetened beverages, eating fast-food, eating fried-food, smoking, and drinking alcohol [15]. Furthermore, some health-related factors are also linked to excess weight (e.g., physical fitness [16,17], blood pressure [6], or sleep quality [18]). Additionally, adolescents with excess weight show consistent higher heightened risk of psychological comorbidities including depression, compromised perceived quality of life, anxiety, depression, self-esteem, and behavioral disorders [19].

Adolescence is recognized as one of the stages that exert a crucial role in the development and maintenance of excess weight and its associated co-morbidities into adulthood [20]. Thus, current research demonstrates that the adolescent period is marked by critical changes in body composition, insulin sensitivity, physical activity, sedentary and diet behaviors, and psychological factors that can increase the risk of having excess weight and maintaining it later in life [21]. The strong evidence found in follow-up studies of childhood obesity reveals the need to intervene early in life and change the environment to prevent the spread of the harmful effects of excess weight into adulthood, as well as associated co-morbidities [21].

On the other hand, although further research is needed, the combination of enhanced physical activity and improved nutrition has emerged as a promising strategy against excess weight among young population (e.g., adolescents) [22]. Thus, a variety of multicomponent lifestyle interventions involving strategies for change in diet and/or physical activity and family involvement may improve body mass index in the young population with excess weight [23,24]. However, higher quality trials are needed as the current evidence is of low methodological quality [23,24]. In addition to targeting whole populations in which a large prevalence of young population has excess weight, future actions also need to apply targeted strategies that reach high-risk adolescents within the context of their schools, families, and communities [25]. Thus, this research project will try to provide relevant information to be able to design intervention programs, as well as to implement effective public policies to try to reverse this alarming situation.

Based on the above, the aim of this project is two-fold: first, to obtain cross-sectional and longitudinal data on the excess weight and their potential sociodemographic, environmental, lifestyle, health-related, cognitive, and psychological factors associated among adolescents from the Valle de Ricote (Region of Murcia, Spain) (aged 12–17 years), and second, to examine the association between excess weight and their potential sociodemographic, environmental, lifestyle, health-related, cognitive, and psychological factors associated among this population.

## 2. Materials and Methods

### 2.1. Study Design and Population

A cross-sectional study and follow-up study will be performed in order to obtain information and to monitor the excess weight and its associations with potential sociodemographic, environmental, lifestyle, health-related, psychosocial, and cognitive factors of adolescents in the Valle de Ricote. Data collection will begin during the 2021/2022 academic year and follow-up is planned to be conducted during the 2024/2025 (3 years later). The acquired data will be processed with the aim of examining the possible relationships between the different evaluated variables that could help to design effective strategies for preventing excess weight in adolescents. Similarly, it will also help to determine trends in increases/decreases in the prevalence of excess weight, as well as which factors (sociodemographic, environmental, lifestyle, health-related, psychosocial, and cognitive factors) are associated with these changes.

A total of three secondary schools from the Valle de Ricote (Region of Murcia, Spain) will be assessed for this study. The sample size will be calculated using the following formula: *n* = ${\left(Z\right)}^{2}\left(\frac{p\;\left(1-p\right)}{e^{2}}\right)$, where “*n*” is sample size, *Z* = 1.96 (95% confidence interval), *p* = prevalence of overweight and obesity (40.0%) [8], and *e* = margin of error (3%), the minimum sample size (by considering 10% non-response rate) will be 1138. This study will involve adolescents using a simple random sampling technique.

Regarding participation in this research project, the parents or legal guardians of the adolescents will receive a written informed consent form to be signed previously. Additionally, both parents or legal guardians and their children will receive an information sheet explaining the aims of this research project, as well as all the tests and questionnaires that will be administered. Likewise, adolescents will also be asked about their willingness to participate in the study. The entire data collection process will be carried out during Physical Education classes. Due to the large amount of information analyzed in this study, data collection (e.g., physical tests, physical examinations, questionnaires) will be carried out in different Physical Education classes.

As inclusion criteria, participants must comply with the following conditions: (1) aged between 12 and 17 years old, (2) registered and/or lived in the Valle de Ricote. Regarding exclusion criteria, adolescents will not be enrolled if they: (1) are exempt from the subject of Physical Education at school, since both the tests and the fulfilment of the questionnaires will be perform during the Physical Education lessons, (2) suffer any pathology that contraindicate physical activity or that demand special attention, (3) are under some kind of pharmacological treatment, (4) are not authorized by the parents or legal guardians to participate in the research project, or (5) do not agree to take part in the research project.

This research project obtained ethics approval from the Bioethics Committee of the University of Murcia (ID 2218/2018) and the Ethics Committee of the Albacete University Hospital Complex and the Albacete Integrated Care Management (ID 2021-85). It will be carried out following the Helsinki Declaration, respecting the human rights of the participants enrolled.

### 2.2. Procedures

A summary with all the variables that were analyzed in the present research project is presented in Figure 1.

#### 2.2.1. Excess Weight Measurements

The body weight of the adolescents will be measured using an electronic scale (with an accuracy of 0.1 kg) (Tanita BC-545, Tokyo, Japan), while height was determined by a portable height rod with an accuracy of 0.1 cm (Leicester Tanita HR 001, Tokyo, Japan). Body mass index (BMI) will be calculated by dividing body weight (in kg) by the height (in squared meters). Furthermore, BMI z-score will be computed by the WHO age-specific and sex-specific thresholds [26], as well as the International Obesity Task Force Criteria [27]. Subsequently, the BMI z-scores obtained will be used to determine excess weight. Waist circumference will be measured to the nearest 0.1 cm at the level of the umbilicus, using a constant tension tape. Moreover, the waist-to-height ratio (WHtR) will be calculated and, therefore, a value ≥0.5 used as a cut-off point to establish abdominal obesity [28]. Skinfold measurements to the nearest 0.2 mm were taken with calibrated steel calipers (Holtain, Crosswell, Crymych, UK) at the triceps and medial calf. These procedures will be based on the recommendations of the International Society for the Advancement of Kinanthropometry (ISAK). Both triceps and medial calf skinfolds were proved to be successful in estimating body fat percentage in children and adolescents [29].

#### 2.2.2. Sociodemographic Factors

Sex and birth date were self-reported. Age was calculated from birth dates. The type of schooling was divided into two categories: (a) public and (b) private with public funds. Area of residence was divided into: (a) urban (>5000 inhabitants) and (b) rural (≤5000 inhabitants) [28]. Students who met at least one of the following conditions were considered to be immigrant: (a) immigrant parents, (b) born outside Spain, or (c) at least one parent from another country. Socioeconomic status (SES) will be assessed by the Family Affluence Scale (FAS-III) [30]. The FAS-III score will be calculated by the sum of the responses from 6 different items: (a) “Does your family own a car, van, or truck?” (0 = no, 1 = yes, one, 2 = yes, two or more), (b) “Do you have your own bedroom for yourself?” (0 = no, 1 = yes), (c) “How many computers do your family own (including laptops and tablets, not including game consoles and smartphones)?” (0 = none, 1 = one, 2 = two, 3 = more than two), (d) “How many bathrooms (room with a bath/shower or both) are in your home?” (0 = none, 1 = one, 2 = two, 3 = more than two), (e) “Does your family have a dishwasher at home?” (0 = no, 1 = yes), (f) “How many times did you and your family travel out of Spain for a holiday/vacation last year?” (0 = not at all, 1 = once, 2 = twice, 3 = more than twice). The final score ranges from 0 to 13 points. Thus, three different categories will be established: (a) low SES (0–2 points), (b) medium SES (3–5 points), (c) high SES (≥6 points). Adolescents will be asked about the educational level of their father/mother/legal guardian individually. Possible choices will be: (a) incomplete primary education, (b) complete primary education, (c) incomplete secondary education, (d) complete secondary education, (e) incomplete higher education, or (f) complete higher education.

#### 2.2.3. Environmental Factors

Home environment

To assess the home environment, adolescents will be asked several questions related to the home environment. Information about the number of people living in their households, as well as the number of siblings, will be solicited. Similarly, the type of family will be asked. The possible options will be: (a) two-parent family, (b) reconstituted/compound family, (c) same-sex family, (d) single-parent family (men), (e) single-parent family (women), (f) extended family, or (g) adoptive family. Furthermore, they were asked for the presence of TV in the bedroom (yes/no).

Barriers for the physical activity practice

To assess the barriers to the physical activity practice, a short scale of perception of barriers to physical activity in adolescents will be applied, which was previously validated among Spanish population [31]. The scale contains 4 different dimensions: incompatibility barriers (2 items), self-concept barriers (4 items), amotivation barriers (4 items), and social barriers (2 items). Each item is scored with a 5-point Likert-type response scale, from 1 (strongly disagree) to 5 (strongly agree).

Social support for physical activity

The instrument used to evaluate social support for physical activity was specifically developed for use in the Motorik-Modul (MoMo) study [32]. The social support scale contains two factors related to the two main providers of social support for physical activity (e.g., family and peers): parental support (8 items) and peer support (3 items). The answers are based on a 4-point rating scale, from 1 (not at all) to 4 (very important). The greater the scores, the higher are the perceptions of social support for physical activity. Evidence of the validity and reliability of this social support scale has been previously indicated [33]. Similarly, this instrument was used in a previous study among Spanish adolescents [34].

Physical environment for physical activity

The physical environment for physical activity will be also assessed by the instrument specifically developed for using in the above-mentioned MoMo study [32]. The items of the physical environment scale are related to the accessibility of recreation facilities (4 items), convenience (3 items), and safety (4 items). Similarly, as in the social support scale, the answers are based on a 4-point rating scale. Higher scores mean a more friendly environment for physical activity. Evidence on validity and reliability of this physical environment scale has been pointed out [33]. This instrument was also used in study performed among Spanish youths [34].

Dog ownership

Participants will be asked if they own a dog and, if so, if they walk the dog. Possible answers are: (a) never/rarely, (b) less than once/week, (c) 1–2 times/week, 3–4 times/week, (d) 5–6 times/week, (e) daily [35]. Participants will be assigned to one of three categories: non-dog-owners, non-dog- walkers, and dog walkers [36].

#### 2.2.4. Lifestyle Factors

##### Movement Behaviors

24 h movement behaviors (accelerometry)

The 24 h movement behaviors (physical activity, sedentary time, and sleep) will be evaluated by accelerometry [37], with ActiGraph (wGT3X-BT) accelerometers. Accelerometers will be placed on the hip as it allows to detect the body’s acceleration/deacceleration during locomotion [38]. Adolescents will be asked to wear the accelerometer uninterruptedly (including sleeping and while engaging in water-based activities (e.g., swimming, bathing) for a minimum of 5 days to obtain 3 full days (3 × 24-h period) of data. This will provide data on total physical activity, light intensity physical activity, moderate-to-vigorous intensity physical activity, total sitting time, and total lying time.

24 h movement behaviors (self-reported)

Physical activity

The Youth Activity Profile Physical (YAP), a 15-item self-report instrument, will be used to obtain information related to physical activity and sedentary behavior among adolescents [32]. The YAP is a self-administered 7-day (previous week) recall questionnaire appropriate for being used in young people aged 8–17 years. The items use a 5-point Likert scale and are separated into 3 sections: (1) activity at school, (2) activity out-of-school, and (3) sedentary habits [39]. Activity at school includes transportation to and from school, as well as activity during Physical Education classes, lunch, and recess time. Activity out-of-school section contains activity before school, activity right after-school, activity during the evening, and activity in each weekend day (Saturday and Sunday). Sedentary habits section is referred to as time spent watching television (TV), playing videogames, using the computer, using a cell phone, and an overall sedentary time item. The Spanish version of YAP (YAP-S) has been validated and adapted previously [40]. From YAP score, relative time spent in sedentary behavior (%/day) and moderate-to-vigorous physical activity (%/day) will be computed [41]. Therefore, adolescents will be categorized into “not meeting physical activity guidelines” (at least an average of 60 min/day of moderate-to-vigorous intensity), and “meeting the physical activity guidelines” (less than an average of 60 min/day of moderate-to-vigorous intensity), according to the WHO guidelines on physical activity for children and adolescents [42].

Recreational screen time

Recreational screen time will be assessed by asking adolescents to declare the time that they spent in different sedentary screen-based pursuits. The following questions will be asked for weekdays and weekends: (a) “How many hours a day, in your free time, do you usually spend watching TV, videos (including YouTube or similar services), DVDs, and other entertainment on a screen?”, (b) “How many hours a day, in your free time, do you usually spend playing games on a computer, games console, tablet, smartphone or other electronic device (not including moving or fitness games)?”, and (c) “How many hours a day, in your free time, do you usually spend using electronic devices such as computers, tablets or smartphones for other purposes (e.g., homework, emailing, tweeting, Facebook, chatting, surfing the internet)?”. A weighted sum of the three questions will be calculated (i.e., 5 weekdays and 2 weekend days). Recreational screen time will be divided into “not meeting screen time guidelines” (>2 h/day), and “meeting the screen time guidelines”. This categorization is according to the Canadian guidelines on screen time for young population [4].

Sleep duration and *siesta* habits

Sleep duration will be evaluated by asking respondents for weekdays and weekend days separately: “What time does your child usually go to bed?” and “What time does your child usually get up?”. The average daily sleep duration will be computed for each participant as follows: [(average nocturnal sleep duration on weekdays × 5) + (average nocturnal sleep duration on weekends × 2)]/7. In addition, two ad hoc questions on *siesta* habits were asked: (1) “Do you usually take a *siesta*?”, with yes or no options, and (2) “How long do you take a *siesta*?”, with answers ranging from: (a) 0–15 min, (b) 15–30 min, (c) 30–45 min, (d) 45–60 min, (e) 60–75 min, (f) 75–90 min, (g) 105–120 min, or (h) 120 or more min. A *siesta* is a traditional short nap taken in the early afternoon, often after the midday meal. Such a period of sleep is a common tradition in some countries, particularly those where the weather is warm (e.g., Mediterranean region). The school timetable for all participants is from 08:15 or 08:30 a.m. to 14:15 or 14:30 p.m., which makes it possible to take a *siesta*. Thus, responses within the range of 9–11 h for 12–13-year-olds and 8–10 h for 14–17-year-olds will be categorized as “meeting sleep guidelines,” and participants out of these ranges will be classified as “not meeting sleep guidelines”, following the National Sleep Foundation’s sleep duration recommendations among the young population [43].

Active commuting to and from school

The mode of commuting to and from school will be measured using a questionnaire for children and adolescents developed through the project “PACO: *pedalea y anda al colegio*” (http://profith.ugr.es/paco, access date is 1 August 2021), which has been validated in Spanish young population [44]. It is a 4-item self-report instrument created to assess the usual mode and weekly frequency of mode of commuting in young population. The questionnaire includes 4 items: (1) “How do you usually get to school?”, (2) “How do you usually get home from school?”, (3) “How did you get to school each day?”, and (4) “How did you get home from school each day?”, answer options were: (a) walk, (b) bicycle, (c) car, (d) motorcycle, (e) bus, (f) public bus, (g) metro/train/tram, or (h) other (the mode description was required). The percentage of usual cycling and active modes of commuting to and from school, and the weekly number of cycling and active travels to and from school will be determined. Furthermore, the objective measure of the commuting distance from school to home for each adolescent will be estimated using Google Maps™ software. Distances will be calculated choosing the shortest walking route from the school postal address to family postal address from each adolescent. The objective time necessary for the journey will be described by Google Maps™ according to the time required for each adolescent to cover the shortest walking route from school to home.

Perceived physical literacy

Perceived physical literacy, understood as a specific intelligence related to physical competence, confidence, knowledge, motivation, and understanding to value and take responsibility for maintaining certain physical activities and pursuits throughout life [45], will be evaluated by the “Perceived Physical Literacy Instrument” (PPLI), which was originally designed for Physical Education teachers [46]. The PPLI includes 9 items scored on a 5-point Likert scale, from 1 (strongly disagree) to 5 (strongly agree). The 9 items of the PPLI are equally divided into three subscales that includes: “knowledge and understanding” (3 items), “self-expression and communication with others” (3 items), and “sense of self and self-confidence” (3 items). Since the original version is quite generic and not created for a designated population or profession, it has been previously used without changes in vocabulary for adolescents [47], showing confirmatory factor analysis a good fit to the model among adolescents [47]. Thus, in the absence of any validation study in the Spanish population, a cultural adaptation process will be developed following the methodology suggested by Beaton et al. [48].

Muscle-strengthening activities

Information about muscle-strengthening activities will be obtained by the following question: “In the past week, how many days did you engage in exercise to strengthen or tone the muscle, such as push-ups, sit-ups, or lifting weights?”. The possible responses are: (a) none, (b) 1 day, (c) 2 days, (d) 3 days, (e) 4 days, (f) 5 days, (g) 6 days, (h) 7 days. This question was used for health behavior surveillance in Canada [49], China [50], United States [51], as well as other countries [52,53]. Likewise, this question was proved to have acceptable reliability for the young population (α > 0.55) [54]. Adolescents who self-reported engaging in muscle-strengthening activities for ≥3 days in the last week will be considered as meeting the WHO recommendation [42].

Organized sport activities

Several ad-hoc questions about organized sport activities will be answered by the adolescents. (1) “Do you attend a sport club?” (yes/no), (2) “What type of sport do you play in that club?”, (3) “How many days do you attend weekly?” (Options range from 0 to 7 days), (4) “How much time per session do you attend?”, and (5) “What is your interest in sport practice?”.

Use of social networks

The use of social networks (Facebook, Twitter, WhatsApp, Instagram, Snapchat, TikTok) will be evaluated by a single item scale asking adolescents what type of social network they use, based on five possible responses: (a) I never or rarely use them, (b) I am a low consumer, (c) I am a medium consumer, (d) I am a fairly high consumer, or (e) I am a very high consumer [55].

##### Eating Patterns

Adherence to the Mediterranean diet

To evaluate the adherence to the Mediterranean diet, the Mediterranean Diet Quality Index for Children and Teenagers (KIDMED) index will be applied [56]. This test was previously validated [56] and widely used in the Spanish young population [57]. The KIDMED index ranges from 0 to 12 and is based on a 16-question test. Items reporting unhealthy characteristics related to the Mediterranean diet are scored with −1 point, and those reporting healthy characteristics with +1 point. The sum of all scores from the KIDMED test will be used to categorize into 3 different levels: (a) optimal Mediterranean diet (>8 points), (2) improvement needed to adjust intake to Mediterranean patterns (from 4 to 7 points), (c) very low diet quality (≤3 points) [56].

Food consumption and energy and nutrient intake

Food consumption and energy and nutrient intake will be estimated by a food frequency questionnaire (FFQ) self-administered, which was previously validated among the Spanish population [58]. This FFQ contains 45 items separated into 12 different food groups: (a) red and processed meat, (b) poultry, fish, and eggs, (c) fruits (fruit, preserved fruit), (d) vegetables (salads and vegetables), (e) dairy products, (f) salted cereals (breakfast cereals, bread, pasta, and rice), (g) sweet cereals (biscuits, pastries), (h) legumes, (i) nuts, (j) sweets (sugar and chocolates), (k) sweetened beverages, (l) alcoholic drinks. Adolescents will be asked for the weekly/monthly consumption of these foods and the weekly average ration of these groups will be computed. Similarly, total energy and nutrient intake will be estimated. Additionally, an ad-hoc question about water consumption will be asked: “How many glasses of water (approximately 200 mL) do you drink per day?”. It will be established if dietary guidelines of the Spanish Society of Community Nutrition [59] are being met or not.

Emotional eating

Emotional eating will be evaluated by the Emotional Eating scale (EES-C) adapted for the young population (aged 8–17 years) [60], which is a 25-item self-report questionnaire scored on a 5-point Likert scale, from 1 (no desire to eat) to 5 (very strong desire to eat). This scale was created to assess the urge to eat in order to deal with negative effects. The Spanish version has been previously adapted and validated for adolescents [61].

Food insecurity

Food insecurity will be determined using the Spanish Child Food Security Survey Module (CFSSM-S) [62]. The CFSSM-S focuses on participants’ perceptions about food insecurity in the households, as well as their worries related to running out of food, eating only cheap foods, not being able to eat a balanced diet, eating less, cutting down on portions of food, skipping meals, going hungry, and not eating for a whole day. This instrument includes 9 items based on a 3-point-Likert scale. Affirmative responses (i.e., “a lot” and “sometimes”) are scored 1 point, while negative responses (i.e., “never”) are scored 0 points. The food security classification will be on the basis of the original study [63] and the US Department of Agriculture’s nomenclature [64]. Thus, 3 different groups will be established: (a) food secure (0–1 points), (b) low food security (2–5 points), and (c) very low food security (6–9 points).

Social Eating Behavior

Social eating behavior will be evaluated by three different statements: “I enjoy sitting down with family or friends and eating a meal together,” “It is important to sit down and eat at least one meal a day with other people (family or friends),” and “I usually eat dinner with other people”. Response options are “strongly disagree”, “somewhat disagree”, “somewhat agree”, or “strongly agree.” These three items will be summed to compute a Social Eating Score (range 3 to 12, Cronbach’s α = 0.70) [65].

Family meals

Frequency of family meals was assessed with the question “During the past 7 days, how many times did all, or most, of your family living in your house eat a meal together?” Response categories are: (a) none, (b) 1 day, (c) 2 days, (d) 3 days, (e) 4 days, (f) 5 days, (g) 6 days, (h) 7 days. To compare adolescents according to whether they were eating meals with their families regularly (i.e., most days weekly), answers will be categorized into “less than 5 meals per week” and “5 or more meals per week” [66].

Meal attitudes

Some meal attitudes (e.g., frequency of social eating, enjoyment of social eating, perceived importance of social eating, perceived importance of meal regularity, eating on the run, and perceived time constraints on meals) will be assessed. Frequency of social eating will be assessed by the item: “I usually eat dinner with other people”. Enjoyment of social eating will be evaluated by the statement: “I enjoy sitting down with family or friends and eating a meal together”. Perceived importance of social eating will be evaluated by the following item: “It is important to sit down and eat at least one meal a day with other people (family or friends)”. Perceived importance of meal regularity will be determined by this statement: “Regular meals are important to me”. Eating on the run was assessed through the item: “I tend to eat on the run”. Perceived time constraints on meals were examined based on this statement: “It is hard to find time to sit down and eat a meal”. For all of these questions, possible response choices are: (a) strongly disagree, (b) somewhat disagree, (c) somewhat agree, or (d) strongly agree. For analyses examining associations of meal attitudes with characteristics of adolescents and their dietary intake, responses will be dichotomized (disagree or agree) [67].

Meal structure

Meal structure will be assessed by four different statements: (1) “It is hard to find time to sit down and eat a meal”, (2) “I tend to eat on the run”, (3) “Regular meals are important to me”, and (4) “I eat meals at about the same time every day”. Response options are: (a) strongly disagree, (b) somewhat disagree, (c) somewhat agree, or (d) strongly agree [68]. These four statements will be summed to form a Meal Structure Score. Both and second response options will be reversed. Thus, a higher score indicates greater meal structure (range 4 to 16) [65].

Meal frequency

The meal frequency of breakfast/lunch/dinner will be examined by the following question: “During the past week, how many days did you eat breakfast/lunch/dinner?”. Response options are: (a) never, (b) 1 to 2 days, (c) 3 to 4 days, (d) 5 to 6 days, and (e) every day [68,69]. The different response options about the number of meals will be scored with 0, 1.5, 3.5, 5.5, or 7 points, respectively, to allow for comparison of mean meal frequencies [65].

Meal duration

To assess meal duration, adolescents will be enquired about how long (on average) breakfast/lunch/dinner typically lasts. Possible options are: (a) I do not eat this meal, (b) 5 min, (c) 10 min, (d) 15 min, (e) 30 min, (f) 45 min, (g) 60 min, or (h) more than 60 min.

Food timing

Food timing will be evaluated using a seven-day dietary record including food timing, which will be completed by the adolescents [70]. The median of the dinner time will be used to create two different groups (“early dinner eaters” and “late dinner eaters”). Midpoint of food intake will be defined as an average of the seven days of the midpoint between breakfast and dinner times (first and last meal). Additionally, we will establish the following variables: (a) social jet lag of dinner timing (the difference between dinner timing on weekends and on weekdays), (b) social jet lag of midpoint of food intake (the difference between the midpoint of food intake on weekends and weekdays). Similarly, intraindividual variation variables will also be defined: (a) dinner phase deviation (the standard deviation of the mean of dinner timing), (b) midpoint of food intake phase deviation (the standard deviation of the mean of midpoint of food intake), (c) inter-day phase change (in dinner timing and midpoint of intake) as follows: $\sqrt{{\left(ai-bi\right)}^{2}}$ (*ai* = dinner timing or midpoint of intake on day *i*, *bi* = dinner timing or midpoint of intake on previous day) [71].

Eating distractions

Some eating distractions will be also assessed. Thus, adolescents will be asked about: (1) eating while talking on the phone/sending SMS/emails or using social networks, (2) watching TV during eating lunch/dinner, and (3) eating while standing. The response choices for each of the two measures are: (a) strongly disagree, (b) a little, (c) somewhat, and (d) very important. Subsequently, eating distractions responses will be classified into “disagree” or “agree”.

Involvement in household food tasks

Involvement in household food tasks among adolescents will be evaluated by two different questions: First, the frequency of helping to prepare food for dinner: “In the past week, how many times did you help prepare food for dinner?”. Response options are: (a) never, (b) one or two times, (c) three or four times, (d) five or six times, and (e) seven times. Similarly, the frequency of helping to shop for food will be also assessed: “In the past week, how many times did you help shop for groceries?” Response options are: (a) never, (b) one or two times, (c) three or four times, (d) five or six times, and (e) seven times [72].

Sustainable food practices

Sustainable food practices will be assessed by asking participants about four different sustainable food practices: ‘How important is it to you that your food is produced as organic/not processed/locally grown/not genetically modified? [68]. Response options for each of these four different practices are: (a) not at all, (b) a little, (c) somewhat, and (d) very important. In addition, responses will be dichotomized to represent whether a participant values (“somewhat” or “very important”) or does not value (‘not at all’ or ‘a little’) each practice. Overall value for sustainable diet practices will be defined at each time point by whether or not an individual valued at least two practices [73].

Perceived cooking skills

Perceived adequacy of cooking skills was assessed by asking: “How adequate are your cooking skills?”. Participants could reply with 4 options: (a) very adequate, (b) adequate, (c) inadequate, or (d) very inadequate. Categories will be summarized as “inade- quate” or “adequate” to describe the adequacy of their skills [72].

Tooth brushing

Tooth brushing frequency will be assessed using the following question: “How often do you brush your teeth?” with five levels of response: (a) more than once a day, (b) once a day, (c) at least once a week but not daily, (d) less than once a week, and (e) never [74]. According to the international recommendations for tooth brushing [75], adolescents will be divided into two groups: “tooth brushing at least twice a day”, or “tooth brushing once a day or less”.

Weight control

To identify adolescents trying to lose weight at the time of participating in this study, adolescents will be asked to report whether they are at present on a diet or doing something else to lose weight. Possible responses are: (a) yes, (b) no, but I should lose some weight, (c) no, my weight is fine, and (d) no, because I need to put on weight [74].

##### Drug Consumption

Tobacco smoking

Tobacco smoking will be determined by two different questions [74]. First, the following question will be asked: “Have you ever smoked in your life?”. The response options are: (a) never, (b) once or twice, (c) 3 to 5 times, (d) 6 to 9 times, (e) 10 to 19 times, (f) 20 to 29 times, or (g) 30 times or more. Similarly, the next question was also asked for the adolescents: “How often do you smoke nowadays?”. The possible responses include: (a) I do not smoke, (b) less than once a week, (c) at least more than once a week yet not every day, and (d) every day. In addition, the smoking status of their parents/legal guardians, siblings and their best friend, will be assessed using a single-item question about their siblings, parents/legal guardians, and best friend. The possible options are: (a) daily smoking, (b) no smoking, (c) quit smoking, (d) do not know, (e) do not have/see that person [74]. The response selection of “daily” will be used to classify that person as a daily smoker.

Alcohol consumption

To assess alcohol consumption status, adolescents will answer the following question: “Have you ever drunk alcohol in your life?”. The possible response options are: (a) never, (b) once or twice, (c) 3 to 5 times, (d) 6 to 9 times, (e) 10 to 19 times, (f) 20 to 29 times, (g) 30 times or more. Similarly, participants will indicate how often they had each of these drinks: beer, wine, spirits, mixed drinks, or any alcoholic beverage. The response scale includes five options, which will be converted into days per week: (a) never (0 points), (b) hardly ever (0.10 points), (c) every month (0.25 points), (d) every week (1 point), and (e) every day (7 points). The average value of the five options related to the alcohol consumption will be computed [76]. Similarly, adolescents will be categorized according to their frequency of alcohol consumption into: (a) not users of alcohol (never drank any alcoholic beverage), (b) regular users (drank any alcoholic beverage every week or more often), and (c) irregular users (drank anything alcoholic beverage every month or less frequently) [77].

On the other hand, information about drunkenness will be used as a measure of excessive alcohol consumption. One single-item will be asked for assessing whether adolescents had drunkenness episodes: “Have you ever had enough alcohol to get drunk?”. The response scale comprises five options: (a) never, (b) once, (c) 2–3 times, (d) 4–10 times, and (e) over 10 times. Furthermore, having been drunk once and having been drunk twice or more will be used as cut-off points [78].

Cannabis use

The use of cannabis will be assessed by the following question: “Have you ever had cannabis in your life?”, with different response choices: (a) never, (b) once or twice, (c) 3 to 5 times, (d) 6 to 9 times, (e) 10 to 19 times, (f) 20 to 29 times, (g) 30 times or more [74].

Initiation of drug consumption

Adolescents will be asked for the initiation age for tobacco smoking, alcohol consumption, first binge-drinking episode, and cannabis use. The possible options are: (a) never, (b) before 12 years, (c) 12 years, (d) 13 years, (e) 14 years, (f) 15 years, (g) 16 years, or (h) 17 years [74].

#### 2.2.5. Health-Related Factors

Objective physical fitness

The ALPHA-FIT Test Battery for young population [79] will be used to evaluate physical fitness. This battery contains different tests to evaluate different components of physical fitness that are explained as follows.

Cardiorespiratory fitness

The maximum volume of oxygen consumed will be estimated by performing a maximum incremental field test (20 m Shuttle Run Test). Participants were tested to run between two lines 20 m apart while keeping a pace with the acoustic signals from a speakerphone audio player with Bluetooth technology.

The test begins at 8.50 km/h and is increased by 0.5 km/h each minute, reaching 18.0 km/h in the 20th minute. Participants will be instructed to run in a straight line to pivot on completing the itinerary between the two lines and to follow the pace set by the audio signals. Adolescents will be encouraged to continue running if they were able to during the course of the test. The test will finish when the adolescent does not reach the end of the lines concurrent with the audio signals on two successive times.

Muscular fitness

Handgrip strength test will be used to evaluate upper body muscular strength using a hand dynamometer with an adjustable grip (TKK 5401 Grip D, Takei, Tokyo, Japan). First, adolescents will receive a short tutorial and verbal order about the test. In addition, the dynamometer will be adjusted based on the child’s hand size as suggested. The test will be carried out in the standing position with the elbow extended and the wrist in the neutral position; the adolescents will receive a verbal support to “squeeze as hard as possible” and exert their maximal handgrip strength for at least 2 s. The test will be performed twice (one attempts per each hand), and the highest score of each hand will be selected. The average of the highest scores achieved by each hand will be selected for additional analyses. Furthermore, normalized handgrip strength will be calculated as the average of the left and right and then divided by body weight (in kg).

On the other hand, the standing broad jump test will be used to determine lower body muscular strength. Adolescents will stand behind the starting line, with feet together, and push off strongly and jump forward as far as possible. The distance will be calculated from the take-off line to the position where the back of the heel nearest to the take-off line lands on the floor. The test will be carried out twice, and the highest score will be selected (in cm).

Speed-agility

The 4 × 10 m Shuttle Run Test will be applied to assess speed agility. Two lines, drawn at a distance of 10 m, and two cones, will be placed at the distant line. The participants will run as fast as possible from the starting line. Every time the adolescents cross any of the lines, they pick up (the first time) or exchange (second and third time) a sponge placed behind the lines. This test will end when the adolescents cross the finish line on one foot. Two attempts will be carried out, and the lowest time will be selected (in seconds) [79].

Flexibility

On the other hand, flexibility will be evaluated by the back-saver sit-and-reach test and the sit-and-reach test. The back-saver sit-and-reach test will require the use of the sit-and-reach standardized box [80] with a slide rule attached to the top and the box places against a wall. The adolescents will be required to sit with the untested leg bent at the knee; the tested leg will be placed straight with the foot placed against the box. In the back-saver sit-and-reach test, only one leg is evaluated at a time. The adolescent slowly reaches forward as far as possible and the furthest position the participant could reach (in centimeters) is recorded. The highest score of two attempts for each leg will be recorded. The average distances reached by the right leg, the left leg, and the mean value for the two legs will be used in the analyses. Reference values for back-saver sit-and-reach test previously reported will be used [80]. Furthermore, the sit-and-reach test will use the previously above-mentioned sit-and-reach standard box. Adolescents will be required to sit with knees straight and legs together, and feet placed against the box. The movement done by the adolescents will be the same as for the back-saver sit-and-reach test. The highest score of two attempts will also be recorded. In both tests, a higher score means a greater performance. Reference values for sit-and-reach test have been previously reported [81].

Self-reported physical fitness

Self-reported physical fitness was assessed by the International Fitness Scale (IFIS), which is composed of a 5-point Likert-scale items asking about the adolescents’ perceived global physical fitness, cardiorespiratory fitness, muscular fitness, speed-agility, and flexibility in comparison with their counterparts’ physical fitness (very poor, poor, average, good, and very good) [82]. The IFIS was originally created in English and then culturally adapted and translated (and reverse-translated) to all the languages enrolled in the HELENA study (e.g., Spanish).

Resting blood pressure

Resting blood pressure will be measured using an automated blood pressure monitor with a fittingly sized cuff (Omrom^®^ EVOLV HEM-7600T-E, Health-care Co., Kyoto, Japan). First, adolescents will be seated in a quiet room for 10 min with their feet on the ground and their back supported. Two readings will be taken, with the second blood pressure reading taken 5 min after the first. The average of the two measurements for systolic blood pressure and diastolic blood pressure will be retained. Subsequently, mean arterial pressure will be computed by the following formula: diastolic blood pressure + [0.333 × (systolic blood pressure—diastolic blood pressure)]. The blood pressure categorization will be performed by age-, sex-, and height-specific cut-off points by the European Society ofHypertension guidelines for the management of high blood pressure in children and adolescents [83]. High-normal blood pressure will be considered as systolic blood pressure and/or diastolic blood pressure ≥than 90th percentile but <than the 95th percentile for young people aged 0–15 years. For those aged 16 years and older, a systolic blood pressure between 130–139 and/or a diastolic blood pressure higher than 85–89. Hypertension, percentile hypertension was considered as systolic blood pressure and/or diastolic blood pressure ≥than the 95th percentile for young people aged 0–15 years. For adolescents aged 16 years and older, a systolic/diastolic blood pressure equal or higher than 140/90 was considered.

Resting heart rate

Resting heart rate of adolescents will be measured using the previously mentioned automated blood pressure monitor (Omrom^®^ EVOLV HEM-7600T-E, Health-care Co., Kyoto, Japan), after they rest and remain seated for 10 min. All measurements will be taken three times in a seated position.

Sleep disorders

Sleep disorders will be evaluated by the BEARS (B = Bedtime Issues, E = Excessive Daytime Sleepiness, A = Night Awakenings, R = Regularity and Duration of Sleep, S = Snoring) scale, a screening tool created to screen the most common sleep disorders in the young population (aged 2–18 years) in the context of a clinical interview [84]. This instrument contains questions that evaluate sleep-related areas such as bedtime problems (e.g., difficulties for going to bed and falling asleep), excessive daytime sleepiness (e.g., behaviors usually related to somnolence during the day), awakening during the night, regularity and duration of sleep, and snoring. A previous study showed the concurrent validity of the Spanish translation of the BEARS to screen sleep disorders in pediatric evaluations [85].

Chronotype

Adolescents’ chronotype will be evaluated by the Morningness/Eveningness Scale in Children (MESC) [86]. The MESC assess chronotype by items relate to adolescent’ preferences toward morning or evening using ratings of the preferred timing of events and how they can perform scheduled activities based on specific times of the day. The scale includes 10 items with 4–5 response choices for each item. A scenario will be shown for adolescents and they have to identify the sentence that best fits them with a score ranging from 1–4 or 5 points (e.g., “Gym class is set for 7:00 in the morning. How do you think you will do?”, with the following answer options: (a) my best, (b) okay, (c) worse than usual, (d) awful). Scores varied from 10 (eveningness) to 42 (morningness). The cut-off points of 18/30 points (which matches the 10th and 90th percentiles, respectively) are usually applied to distinguish the prototypical morning, intermediate, and evening types. However, assuming less restrictive cut-off points (MESC values of 20/28), no sex differences have been detected in the proportion of morning, intermediate, and evening types. This scale was previously validated and translated to the Spanish language [87].

Pubertal status

Pubertal status will be assessed according to Tanner stage [88]. Signs of puberty will be scored based on pubic hair status through the standard pictures of pubic hair development from the Tanner scale. The degree of pubertal development will also be assessed based on the size of the scrotum, testes, and penis in boys, and based on the breast development in girls using the standard pictures of Tanner and 5 categories for both sexes.

#### 2.2.6. Cognitive Factors

Academic performance

School records will be provided at the end of the academic year by each high-school. Academic performance will be assessed in two different ways. Firstly, academic performance will be evaluated according to the grade obtained in Language, Mathematics, Language and Mathematics (combined), English, as well as the grade point average of these three subjects. Previous studies have used these measurements as an indicator of academic performance [89,90]. Secondly, academic performance will be assessed by computing the grade point average of all the subjects taken by the adolescents [91].

#### 2.2.7. Psychological Factors

Eating disorders

The risk of having eating disorders will be assessed by the Sick, Control, One, Fat, Food (SCOFF) questionnaire, a five-question test that can be both hetero- and self-administered with an acceptable sensitivity and specificity at a threshold of two (i.e., if patients provided positive responses to at least two of the five questions). It will be administered by two psychologists. The Spanish SCOFF questionnaire version has been validated for its use in primary care settings [92]. To indicate the risk of having an eating disorder, a score ≥2 points will be used [92].

Body image dimension satisfaction

The evaluation of the body image dimension will be analyzed through the Body Image Dimensional Assessment (BIDA) questionnaire [93], which was previously validated in Spanish adolescents [94]. This questionnaire includes 4 items that evaluate the subjective body image, by a scale which ranges from 1.8 to 5.2 points. Different direct indices can be established from the response options: body dissatisfaction (which shows the difference between the actual and the ideal body image), sexual body dissatisfaction (which shows the difference between the present body image and the most attractive figure for the opposite sex), and compared body dissatisfaction (which shows the difference between the present body image and that of most people of the respondent’s age and sex. The aim of the BIDA is to ascertain the extent to which the participant’s body image fits with his/her desired body image. Body dissatisfaction, sexual body dissatisfaction, and compared body dissatisfaction scores can oscillate from −100% to 100%. Positive values mean that the present estimation is greater than the idealized figure, and vice versa for negative values. The BIDA index shows the final result of the scale and ranges from 0 to 100 points, 0 point being the lowest body dissatisfaction and 100 points the highest body dissatisfaction. This index is the product of the average of the absolute values of body dissatisfaction, sexual body dissatisfaction, and compared body dissatisfaction.

Health related quality of life

Health related quality of life (HRQoL) will be assessed by KIDSCREEN-10 [95]. This instrument provides a global index of HRQoL or emotional well-being using 10 items that includes physical, psychological, and social aspects of well-being. Adolescents should indicate whether (during the last week) they feel: (1) good and fit, (2) full of energy, (3) sad, (4) lonely, (5) if they have enough time for themselves, (6) if they are able to do the things they want in their free time, (7) if their parents treat them fairly, (8) if they have fun with their friends, (9) if they do well in the school, and (10) if they are able to concentrate or pay attention. Each item is scored on a scale of 1 to 5 points, which correspond to the following options: (a) never, (b) almost never, (c) sometimes, (d) almost always, and (e) always. These response options will be applied to almost all items (with the exception of item 1 and item 9). Thus, options for items 1 and 9 are as follows: (a) not at all, (b) slightly, (c) moderately, (d) very, and (e) extremely. In addition, the scores of items 3 and 4 are reversed. The sum of all the scores of KIDSCREEN-10, which varies from 10 to 50, indicates the adolescents’ HRQoL. The greater the score, the higher the HRQoL. The internal consistency of the KIDSCREEN-10 Spanish version has been found to be acceptable (α < 0.70) among the young population [96].

Self-esteem

The Rosenberg’s Self-Esteem Scale [97] will be used to estimate global self-esteem. This scale includes 10 items ranked on a 4-point Likert scale, which varies from 1 (not at all true for me) to 4 (very true for me). The total score ranges from 10 to 40 points, and greater scores denote a higher self-esteem level. The reliability of the Rosenberg’s Self-Esteem Scale has shown its initial validation in relation to adolescents [97]. The translated and validated Spanish version of the Rosenberg’s Self-Esteem Scale is being used, which has shown satisfactory levels of internal consistency and temporal stability [98].

Self-regulation

Self-regulation, understood as the capacity to control and plan their own behavior in a flexible manner, based on the intended outcomes, will be assessed by the Spanish Short Self-Regulation Questionnaire (SSRQ) [99]. This use of this questionnaire in adolescent samples has been previously recommended [99]. The original Self-Regulation Questionnaire (SRQ) was developed by Brown et al. [100]. From the original version, the SSRQ includes a reduced structure of 17 items categorized into 4 different dimensions (goal-setting, perseverance, decision-making, and learning from mistakes), with a 5-point Likert scale scored from 1 (none) to 5 (a lot) [101].

Self-efficacy

To asses adolescents’ self-efficacy, the Spanish version of the General Self-Efficacy Scale will be applied [102]. This scale showed a great internal consistency among Spanish adolescents (α = 0.89) [103]. The original General Self-Efficacy Scale [104] was created to evaluate the stable feeling of personal competence to cope successfully with a broad range of stressful circumstances. Despite the original response format of the Spanish version of the General Self-Efficacy Scale, a 10-point Likert-type response scale will be used [105]. This includes 10 items (e.g., “I can find a way to get what I want even if someone opposes me”) and varies from 10 to 100 points.

Psychosocial health

Psychosocial health will be assessed by the one-sided self-rated version of the Strengths and Difficulties Questionnaire (SDQ) [106], which is widely used to assess different behavioral, emotional, and social problems related to psychosocial health in the young population (aged 11–17 years). In this study, the Spanish version of the SDQ (www.sdqinfo.org, access date is 1 August 2021) was used, which was validated in previous studies [107,108]. The SDQ contains 25 items divided into 5 different subscales: (a) emotional problems, (b) conduct problems, (c) hyperactivity, (d) peer problems, and (e) prosocial behavior. A 3-point Likert scale with different options was used: 0 = not true, 1 = somewhat true, or 2 = certainly true. The score for each subscale varies from 0 to 10 points. The first four subscales (i.e., emotional problems, conduct problems, hyperactivity, and peer problems) will be considered to determine a global psychosocial problems score. The fifth subscale (i.e., prosocial behavior) evaluates resources rather than problems, being conceptually different from the evaluation of psychosocial problems. For this reason, its score will not be included in the total psychosocial problems score [106]. In addition, internalizing and externalizing problems will be computed. The internalizing score ranges from 0 to 20 and is the sum of the emotional and peer problems scales. The externalizing score ranges from 0 to 20 and is the sum of the conduct and hyperactivity scales.

Satisfaction with life

Satisfaction with life will be assessed by the Satisfaction with Life Scale (SWLS) [109], a brief five-item instrument that evaluates the general satisfaction that the individual has with his or her life, understanding that a higher score reflects greater satisfaction. It includes Likert scale of 7, from 1 as “strongly disagree” and 7 as “strongly agree”, with scores ranging from 5 to 35 points. For this research, the adapted and translated Spanish version will be used [110] and the normative data previously proposed to classify life satisfaction as follows: (a) very dissatisfied (5–9 points), (b) dissatisfied (10–14 points), (c) somewhat dissatisfied (15–19 points), (d) neutral (20 points), (e) somewhat satisfied (21–25 points), (f) satisfied (26–30 points), and (g) very satisfied (31–35 points) [111].

Positive and negative affect

Positive and negative affect will be evaluated by the Positive and Negative Affect Schedule 10-Children (PANAS-C10) [112], which is a well-validated brief measure of mood for youth. For this research, the PANAS-C10 Spanish version will be used [113]. The PANAS-C10 consists of five negative affect items (mad, miserable, scared, sad, afraid) and five positive affect items (happy, proud, joyful, lively, cheerful). Adolescents are asked to specify to what degree they have felt each emotion in the past few weeks, with items rated on a 5-point Likert-type scale, ranging from 1 (very slightly or not at all) to 5 (extremely). Both positive and negative affect items are summed separately, in two different subscales. The higher the scores in positive or negative scale, the greater positive and negative effects, respectively.

Bullying and Cyberbullying

Bullying and cyberbullying, from the point of view of the bullies and the victims, will be evaluated using the European Bullying Intervention Project Questionnaire (EBIPQ) [114] and the European Cyberbullying Intervention Project Questionnaire (ECIPQ) [115], respectively. The EBIPQ contains 14 items, 7 for victimization and 7 for abuse, covering specific behavior such as direct physical abuse (e.g., “someone has hit me”), indirect abuse (e.g., “someone has spread rumors about me”), verbal abuse (e.g., “someone has insulted me”), psychological abuse (e.g., “someone has threatened me”), and social exclusion (e.g., “I have been excluded or ignored by other people”). The ECIPQ includes 22 items, 11 for cyber-abuse and 11 for cyber-victimization, covering specific behavior such as identity theft (e.g., “someone has hacked into my account and pretended to be me”), the uploading and/or altering of embarrassing images or videos (e.g., “someone has posted embarrassing photographs or videos of me on the Internet”), and indirect abuse (e.g., “someone has spread rumors about me on the Internet”). The reliability of both Spanish versions of ECIPQ and EBIPQ to assess bullying and cyberbullying jointly has been previously indicated [116]. Similarly, the Youth Outcome Measures for AfterSchool KidzLit Evaluation will be used, to obtain information about bullying from the point of view of the bystanders [117]. It is evaluated by 3 statements, (e.g., “You helped someone who was being bothered”), on a 5-point Likert-type scale (0 = never, 4 = many times). This tool has been also previously used in adolescents [118].

Depression, anxiety, and stress

Emotional symptomatology will be evaluated by the Depression, Anxiety and Stress Scale (DASS-21) [119], which includes 21 items scored on a 3-point Likert-type scale. The score of each item varies from 0 (did not apply to me at all) to 3 (applied to me very much or most of the time), and is divided uniformly in 3 subscales: depression, anxiety, and stress. The Spanish version, which had adequate reliability, will be used in this study [120]. The cut-off points established for the presence of depression, stress, and anxiety will be ≥6 points, ≥5 points, and ≥6 points., respectively. This choice was because of the optimal sensitivity and specificity shown by these cut-off points in adolescents [121].

Experiential avoidance and psychological inflexibility

The Acceptance and Action Questionnaire-II (AAQ-II) [122] will be used as a measure of experiential avoidance and psychological inflexibility. This questionnaire includes 7 items using a 7-point Likert scale. The items denote an unwillingness to experience unwanted emotions and thoughts (e.g., “I am afraid of my feelings”, “I worry about not being able to control my worries and feelings”) and the inability to be in the present moment and behave towards values-directed actions when experiencing psychological events that could undermine them (e.g., “My painful experiences and memories make it difficult for me to live a life that I would value”, “My painful memories prevent me from having a fulfilling life”, “Worries get in the way of my success”). The Spanish version of the AAQ-II has been shown to be valid and reliable to assess experiential avoidance and psychological inflexibility [123].

Addiction to the social networks

The Short Social Networks Addiction Scale-6 Symptoms (SNAddS-6S) [124] will be used to determine addiction to social networks. This tool includes six items related to salience, tolerance, mood modification, relapse, withdrawal, and conflict, as well as a unifactorial structure and has previously been validated among Spanish adolescents [124].

### 2.3. Statistical Analysis

Means (*M*) and standard deviation (*SD*) or frequencies (*n*) and percentages (%) will be reported for all quantitative or qualitative variables, respectively. Data normality will be verified by a Kolmogorov–Smirnov test with Lilliefors correction, as well as the homogeneity of variances by a Levene test. After that, the data will be analyzed using Student’s t-test or Mann–Whitney *U* test for two-groups comparisons, and Kruskal–Wallis *H* test or one-way ANOVA for three group comparisons, depending on the compliance with the normality assumption. In case of differences between groups (variables non-normally distributed), the post-hoc test will be performed by Mann–Whitney *U* test with Bonferroni correction to account for the inflation of type-I error due to multiple comparisons made. Conversely, for variables normally distributed, Tukey’s *HSD* or the Dunnett *T3* test will be used (depending on the homogeneity of the variances). Associations between qualitative variables will be determined using Pearson’s chi-square test. For quantitative variables, the association will be determined through Pearson’s *r* or Spearman’s rho (*ρ*), according to the meeting of the normality assumption. Furthermore, analyses of covariance (ANCOVA) will be used to estimate differences between mean values of obesity-related parameters across different groups established in relation to sociodemographic, environmental, lifestyle, health-related, cognitive, and psychological variables. Furthermore, post-hoc pairwise comparisons (through Bonferroni test) will be performed to verify differences between mean values of obesity-related parameters across established groups. In addition, binary/multinomial logistic regression analyses will be performed to predict the odds of having excess weight in relation to the different sociodemographic, environmental, lifestyle, health-related, cognitive, and psychological factors analyzed. Data analysis will be carried out by the software SPSS (IBM Corp., Armonk, NY, USA) for Windows (version 25.0). A *p*-value ≤ 0.050 was considered to determine statistical significance.

## 3. Discussion

To our knowledge, this will be the first study that provides the specificity, update, and epidemiological data of excess weight and its potential associated factors (sociodemographic, environmental, lifestyle, cognitive and psychological) in adolescents (aged 12–17 years) in the Valle de Ricote (Region of Murcia, Spain).

There are some studies determining excess weight at a national level, such as Estudio ALADINO [125] or Estudio PASOS [126]. However, they do not differentiate between Spanish regions. Moreover, Estudio ALADINO [125] only includes children from 6 to 9 years. Similarly, the last Spanish National Health Survey [8] assesses excess weight in young people by parent-report questionnaires, which could introduce error and bias to the results obtained [127]. Given these limitations, it is necessary to provide information with more objective measures (e.g., BMI, waist circumference, skinfolds), as well as in other specific regions of Spain, to offer better understanding and monitoring of excess weight among adolescents. This justification is reinforced by the alarming proportion of excess weight previously reported in the Valle de Ricote among schoolchildren—as mentioned earlier, it is higher than the average of both the Region of Murcia and Spain as a whole [11].

On the other hand, as reported in the Report of the Commission on Ending Childhood Obesity [128], WHO advises that the monitoring and evaluation of excess weight in young people should serve to increase awareness of this problem and produce advances in the development, implementation, and effectiveness of interventions by addressing vital elements through coordinated and multifactorial actions. Given that no single intervention can halt the progress of the excess weight epidemic, knowing its sociodemographic, environmental, lifestyle, health-related, cognitive, and psychosocial factors could help to design intervention programs/public policies to counteract the deleterious effect of this problem on adolescents’ physical, social, and psychological well-being, as well as the risk of having excess weight and noncommunicable diseases later in life [7].

In addition to the possible above-mentioned benefits from the development of this project, the results obtained will be an opportunity for parents/legal guardians since it will offer information about the situation of their child regarding his/her excess weight status (and other aspects related to their health). Furthermore, knowing more information about excess weight in adolescents and its associated factors may be an important first step to revert this worrying situation in the Region of Murcia. This will allow for more targeted and personalized interventions among this population in the future. At the same time, by developing this research project, it will be possible to establish collaborations with different institutions/companies interested in this project to obtain reference data that can be used to classify and compare future users of the overweight monitoring service. Thus, it will be possible to design intervention programs that meet the needs found through this project and to design novel strategies to prevent and reduce the impact of excess weight, as well as its derived consequences.

On the other hand, the implementation of this project will also help educational centers (headmasters, headmistresses, teachers) to acquire global information on numerous variables of the health status of their students, as well as governmental institutions (Regional Ministries, City Councils, etc.) to obtain valuable health-related data on this population. This helps to raise awareness among students about the importance of maintaining a healthy weight, as well as about the different health variables analyzed in this study. In addition, teachers (especially Physical Education teachers) will be able to carry out specific measures during their classes in order to promote/encourage aspects that will improve the health status of students.

Additionally, the development of this research project will also be beneficial to the Global Matrix initiative [129]. This project tries to offer a better understanding of the global variation in child and youth physical activity and its correlates though different Report Card grades, including (in the last 3.0 edition) a total 49 countries [130]. The next 4.0 edition includes a novel and pioneering aspect such as the analysis of some states/regions in addition to countries. Thus, the Region of Murcia has been included for the first time in this project. Consequently, all the information obtained in this research project in relation to the 10 compulsory indicators analyzed in Global Matrix (e.g., level of physical activity, sedentary behavior, active transportation, physical fitness, organized sport activities, social support for physical activity), as well as other additional indicators that will also be evaluated (sleep duration, overweight), will be available to be reported in the Report Card from the Region of Murcia.

This study is without certain limitations. First, due to the design of this study (e.g., cross-sectional, longitudinal), it will not be possible to conclude that the observed relationships reflect causal associations. Second, this study will not collect blood samples from the adolescents enrolled, which could offer valuable information on several biomarkers related to excess weight. Conversely, this study has several strengths. For instance, it will include a large and representative sample of adolescents from the Valle de Ricote (Region of Murcia). In addition, a large number of variables related to excess weight from a wide range of different domains will be analyzed. Furthermore, the results from this study will be especially meaningful for a better understanding of the factors associated with excess weight in the Region of Murcia (the Spanish region with the highest prevalence of excess weight), as well as for developing effective interventions to prevent excess weight and related chronic diseases.

## 4. Conclusions

This research project will aim to determine the prevalence of excess weight and interrelate their potential sociodemographic, environmental, lifestyle, health-related, cognitive, and psychological factors associated. The obtained results will help to manage and propose possible multidisciplinary interventions and strategies in order to prevent and reduce the high excess weight in adolescents from the Valle de Ricote. In addition, the results of this study will be important to determine whether children and youth of the Valle de Ricote meet with recommended values and are able to undertake the pertinent interventions to attend this situation. Furthermore, orientations will be given to transfer the obtained results to the public sector to evaluate or change the adopted policies.

## Figures and Tables

**Figure 1 children-09-00370-f001:**
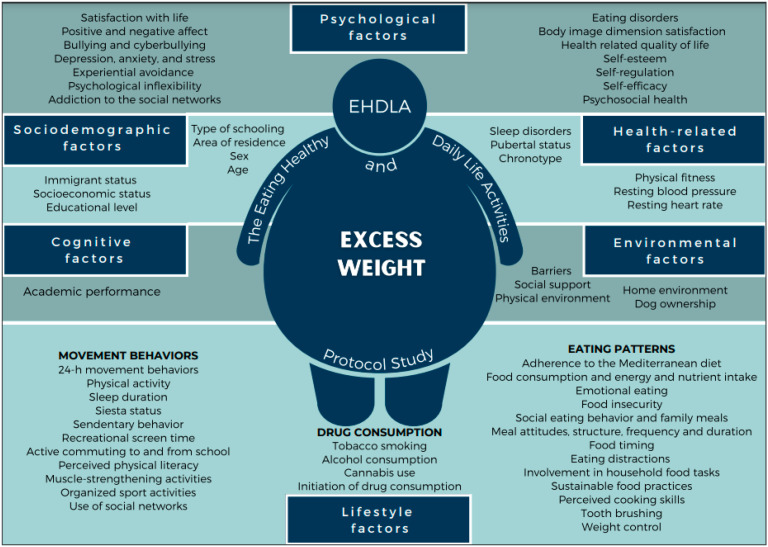
Variables analyzed in the Eating Healthy and Daily Life Activities Study.

## Data Availability

The data presented in this study are available on request from the corresponding author. The data are not publicly available because they belong to minors.

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
