# Peer review of "The Eating Healthy and Daily Life Activities (EHDLA) Study"

_children, 2022, doi:10.3390/children9030370_

Round 1

Reviewer 1 Report

Dear Author,

Thank you for the excellent study protocol. You are planning a huge and important study. I have only a few minor comments to add.
First, I think study title should be expanded as the study will collect information not only on eating and daily activities. Your study seems to be based on biopsychosocial model similarly to HBSC. 
In Section 2.1 first paragraph please explain, how follow-up data will be collected and used? Will the same sample be investigated in the future to collect information on new cases of overweight and obesity? What will be the duration of the follow-up? What information will be collected to link data from both health examinations?

As study is really multidimensional and long, an average duration of filling the questionnaire and taking objective measurements should ne indicated.

Page 6, Screen time: some screen-based behaviours could be done at the same time: e.g., someone can browse social media and watch TV at the same time. Summarizing the duration of these behaviours can overestimate screen-time.
Next paragraph on sleep duration and siesta should incorporate an explanation on siesta as it is not common for all countries. It is not clear, for how long children stay at school and if they all have a possibility for siesta?

Reviewer 2 Report

Thank you for inviting me to review this manuscript, titled “The Eating Healthy and Daily Life Activities (EHDLA) Study: A Protocol Study”.  This study propose to analyze obtain cross-sectional and longitudinal data on the excess weight among adolescents and to evaluate the association between excess weight its relationships with potential sociodemographic, environmental, lifestyle, health-related, cognitive, and psychological factors among this population.

The title extension is adequate.

The specific issue or problem is defined.

The structure is appropriate for this type of work.

The organization of work is adequate.

The main question addressed by the research is relevant and interesting.

The methodology is adequate.

The proposed objective is relevant, as well as the evidence provided and the accompanying discourse.

The paper is presented in an understandable way.

The discussions and conclusions are consistent.

However, the manuscript would benefit from some small suggestions or changes. See specific suggestions below.

It would be necessary to alphabetize the keywords in the abstract unless you want to emphasize some of them

It is not necessary to use a zero before the decimal point when the number cannot be greater than 1. This occurs on page 14 Health related quality of life at the “(α < 0.70)” It is recommended to put (α < .70)

Expand the limitations of the study and some other final paragraph with the educational implications of the present study
